# Critical Role of Hemopexin Mediated Cytoprotection in the Pathophysiology of Sickle Cell Disease

**DOI:** 10.3390/ijms22126408

**Published:** 2021-06-15

**Authors:** Rani Ashouri, Madison Fangman, Alicia Burris, Miriam O. Ezenwa, Diana J. Wilkie, Sylvain Doré

**Affiliations:** 1Department of Anesthesiology, Center for Translational Research in Neurodegenerative Disease, University of Florida College of Medicine, Gainesville, FL 32610, USA; ashourir15@ufl.edu (R.A.); maddiefang@ufl.edu (M.F.); aliciaburris@ufl.edu (A.B.); 2Department of Biobehavioral Nursing Science, University of Florida College of Nursing, Gainesville, FL 32610, USA; moezenwa@ufl.edu (M.O.E.); diwilkie@ufl.edu (D.J.W.); 3Departments of Neurology, Psychiatry, Pharmaceutics, and Neuroscience, McKnight Brain Institute, University of Florida, Gainesville, FL 32610, USA

**Keywords:** heme, heme binding protein, hemoglobinopathy, hemolysis, inflammation, immune modulation, oxidative stress, sickle cell anemia, vaso-occlusion, therapy

## Abstract

Circulating hemopexin is the primary protein responsible for the clearance of heme; therefore, it is a systemic combatant against deleterious inflammation and oxidative stress induced by the presence of free heme. This role of hemopexin is critical in hemolytic pathophysiology. In this review, we outline the current research regarding how the dynamic activity of hemopexin is implicated in sickle cell disease, which is characterized by a pathological aggregation of red blood cells and excessive hemolysis. This pathophysiology leads to symptoms such as acute kidney injury, vaso-occlusion, ischemic stroke, pain crises, and pulmonary hypertension exacerbated by the presence of free heme and hemoglobin. This review includes in vivo studies in mouse, rat, and guinea pig models of sickle cell disease, as well as studies in human samples. In summary, the current research indicates that hemopexin is likely protective against these symptoms and that rectifying depleted hemopexin in patients with sickle cell disease could improve or prevent the symptoms. The data compiled in this review suggest that further preclinical and clinical research should be conducted to uncover pathways of hemopexin in pathological states to evaluate its potential clinical function as both a biomarker and therapy for sickle cell disease and related hemoglobinopathies.

## 1. Introduction

### 1.1. Heme Clearance

Hemopexin (Hpx) is a unique internal scavenging glycoprotein that is responsible for the homeostatic maintenance of blood through the regulatory binding of free heme, which eliminates heme’s harmful pro-oxidant and pro-inflammatory potential [1]. Free hemoglobin (Hb) in circulation is oxidized to methemoglobin (metHb), which rapidly degrades to free heme. After metHb degradation, free heme is ultimately bound by Hpx. This binding can occur before or after oxidation to hemin and is predominantly recognized as a 1:1 binding ratio. In the past few decades, a few studies have unveiled the evidence of multiple heme binding sites on Hpx, indicating a need for a further evaluation of this association ratio [2,3,4]. After Hpx–heme complexes are formed, the elimination of heme is mediated by macrophages, typically via the membrane-bound receptor CD91, also known as LRP1. Via signal transduction, CD91 causes the complex relocation of heme to parenchymal cells in the liver for heme catabolism and Hpx recycling. CD91 is also found on the cell membranes of hepatocytes that serve to recognize Hpx-heme complexes and facilitate the subsequent endocytosis of heme into the liver for further breakdown [5,6]. Hpx exhibits the highest protein-binding affinity in the body for free heme and, consequently, plays a significant role in mediating outcomes after injury or disease (Figure 1). For example, Hpx scavenging of free heme contributes to the protection of numerous cells and regulatory processes, such as macrophages, endothelial cells, and hepatic parenchymal cells, through the alleviation of heme toxicity and the oxidation of lipoproteins [5]. Additionally, Hpx works with haptoglobin (Hp) to simultaneously combat downstream pro-oxidant and pro-inflammatory damage mediated by the presence of free heme and its predecessor, hemoglobin (Hb). Via receptor-mediated signal transduction and direct Fenton oxidation, heme and Hb, as well as their oxidized respective counterparts hemin and metHb, induce cell damage and hemodynamic conditions that are ultimately deleterious to the prognosis of several pathologies [7]. By using Hpx and Hp to clear these compounds from circulation, it is possible to reduce the production of reactive oxygen species (ROS) and downstream oxidative and inflammatory damage. It is well documented that heme is a potent inducer of heme oxygenase-1 (HO1) involving the Nrf2 pathway, and the activation of such pathway leads to the increased expression of various other proteins that can mitigate heme-related toxicity. This protection is critical in pathologies that induce increased hemolytic conditions, such as sickle cell disease (SCD).

### 1.2. Sickle Cell Disease

SCD is caused by mutations in the Hb subunit β (HBB) gene coding for the β-globin subunit of the Hb molecule. Specifically, in the series of hemoglobinopathies included under the umbrella term SCD, at least one of the two β-globin subunits of Hb is replaced with the abnormal β-globin hemoglobin S (HbS). In effect, the mutated β-globin gene leads to polymerized Hb chains in the deoxygenated state. Other mutations of the HBB gene include hemoglobin C, hemoglobin E, or variants leading to underexpressed β-globin subunits. These subsets of the disease with an underexpression of β-globin are referred to as β-thalassemia, which is often comorbid with other hemoglobinopathies. While these pathologies are often seen together, a discussion specifically on Hpx in the case of β-thalassemia was omitted to focus on the hemolytic disorders of SCD. Sickle cell anemia (SCA) specifically refers to homozygosity for the HbS gene, in which patients with SCD produces two abnormal β-globin subunits. In the United States alone, approximately 100,000 Americans are affected, and it occurs in 1 in 1000 to 1400 Hispanic Americans and 1 in 500 African Americans [8,9,10]. Additionally, SCD is common internationally, with more than 300,000 children born with SCD annually, and therefore is of particular interest for therapeutic development [11].

### 1.3. Sickle Cell Disease and Hemolysis

The dynamics of Hpx are crucial to outcomes for patients with SCD. Intracellular Hb polymers induce the sickled shape of erythrocytes, which shortens the lifespan of these cells and leads to occlusion of the vasculature, as well as erythrocyte injury [6]. After erythrocyte injury, there is an excessive degree of extravascular and intravascular hemolysis. The latter leads to an elevated level of free, decompartmentalized Hb and heme. Specifically, in SCD, heme exhibits an increased dissociation constant from sickled Hb, further contributing to high levels of free heme [12]. These processes typically lead to symptoms of pain from vaso-occlusion, including acute lung injury, acute chest injury, pulmonary hypertension, vaso-occlusive crises, and ischemia-reperfusion injury [13]. The accumulation of free heme and Hb, as well as hemin and metHb, initiate numerous harmful pro-oxidant and inflammatory cascades. These inflammatory cascades include toll-like receptor (TLR) 2 and TLR4, which initiate the broad innate immune response due to their recognition of damage-associated molecular patterns (DAMPs) [14,15,16]. These response cascades ultimately lead to a decreased quality of life for patients with SCD because they cause increased severe acute lung injury, vaso-occlusion, and pain crises, among other conditions. It has been reported that hyperalgesia, one of the most prominent chronic manifestations of SCD, is directly exacerbated by TLR4 activation via neuroinflammation, microglial activation, and endoplasmic reticulum stress [13]. This evidence has been corroborated by analyses of patients with SCD showing increased amounts of free heme in circulation [17]. In reference to transfusion therapies in SCD patients, reducing the activation of the innate immune response against free heme by utilizing Hpx could hypothetically mitigate the dependence on immunosuppressive therapies. Clinically, this approach is dynamic and must be further studied before application. Additionally, the altered scavenging of vasodilating nitric oxide (NO) by free Hb could lead to worsened vasoconstriction. With narrowed systemic vasculature and pathologically sticky erythrocytes, patients are prone to painful and deleterious vaso-occlusion (Figure 2).

In addition to the pain and pulmonary symptoms associated with sickled cells resulting from TLR4 activation, patients with SCD experience an increased incidence of ischemic events. Most likely due to the compact vasculature of the brain in combination with the increased stickiness of sickled erythrocytes, ischemic pathologies are predominantly seen in children and the elderly [10,18,19]. It is well-documented that Hpx plays a neuroprotective role after ischemic injury; therefore, it should be evaluated as a dynamic therapeutic target, specifically in patients with SCD [20,21,22,23]. According to Li et al., the local expression of Hpx by neurons contributes to protection from free heme through the induction of HO isoenzymes, namely HO1 [20]. The upregulation of HO1 in astrocytes and fibroblasts in the ischemic penumbra exhibit protective effects against ischemic insults by reducing oxidative stress and apoptosis. In Hpx^−/−^ mice, there is evidence of increased oxidative damage, infarct volume, and overall neurological deficiency after ischemic stroke [20,23]. This finding indicates that Hpx plays a multifaceted role in neuroprotection against ischemic events commonly associated with SCD. Here, we summarize the current understanding of Hpx dynamics in sickle cell-associated disorders, specifically those worsened by the presence of free heme. Table 1 and Table 2 summarize preclinical and clinical studies relating to the role of Hpx in SCD pathophysiology, respectively.

## 2. Search Terms and Strategy

The papers used in this review were retrieved through extensive searches in four research databases: PubMed, Google Scholar, Embase, and OneSearch (the University of Florida’s library database). Relevant articles from 1988 to the present were used, with a primary focus on research published within the last 5 years. Only articles published in English were selected for this review. The following search terms were used in all four databases: “hemopexin AND (sickle cell disease),” “hemopexin AND (sickle cell anemia),” “haemopexin AND (sickle cell disease),” “(beta-1B-glycoprotein) AND (sickle cell disease),” and “(beta-1B-glycoprotein) AND (sickle cell anemia).” Additionally, abbreviations were used in place of sickle cell disease (SCD) and hemopexin (Hpx) to perform a more comprehensive search. To mitigate potential bias, each author conducted the search independently, and the list of resulting papers was cross-referenced.

## 3. Immune Modulation

### 3.1. Hemopexin Mediation of Pro-Inflammatory Cytokine Expression

A large percentage of SCD pathophysiology is influenced by Hpx via its immunomodulatory effects. These effects occur independently of any changes in plasma heme or Hb. This finding suggests that there is a mechanism by which Hpx, which is often depleted in patients with SCD, can alter expression or activation of pro-inflammatory, clinically deleterious cytokines and transcription factors. Belcher et al. reported in a preclinical mouse model of Townes-SS mice that the co-infusion of Hpx with Hb led to significantly lower values of phospho-p65 NF–кB (*p* < 0.01) [6]. Furthermore, they reported that the co-infusion of Hpx and Hb reduced the expression of downstream NF–кB transcription target genes, vascular cell adhesion molecule 1 (VCAM-1), and intracellular adhesion molecule (ICAM). The simultaneous reduction of these adhesion molecules minimizes the risk of occlusion and stasis by maintaining vascular integrity. Additionally, Hpx-treated mice showed a lower expression of chemokine ligand 5 (CCL5), which is critical to pro-inflammatory leukocyte recruitment, than control mice (*p* < 0.01) [6]. Vinchi et al. similarly showed that Hpx-deficient mice challenged with exogenous heme infusions had a significantly higher expression of pro-inflammatory interleukin (IL)-6 in splenic myeloid CD11b+ cells [1]. These cells included granulocytes, monocytes, and ferromagnetic macrophages. The authors also used in vitro and in vivo experiments to show that Hpx diminished heme-induced reticuloendothelial system macrophage phenotype switching to the pro-inflammatory M1 subset. In short, the M1, or “classically activated” macrophage phenotype, potentiates activation of the immune and inflammatory responses while the M2, or “alternatively activated” phenotype, functions in immunoregulation and tissue remodeling or repair. In the in vitro studies, the authors used bone-marrow derived macrophages (BMDMs). They first exposed BMDM sample groups to aged red blood cells, hemin, or iron-nitrilotriacetate (FeNTA), then assessed for markers of M1 or M2 polarization by using qRT-PCR. Exposure to these heme or iron sources led to increased M1 polarization, including of the markers eMHCII, CD86, CD14, TNFα, IL-6, and IL-1β, to a varying degree and significance per treatment. The expression of M2-indicating markers CD206, IL-10, and Arginase-1 was reduced. The authors conducted an equivalent in vivo experiment to evaluate macrophage phenotype switching. Mice were injected intravenously with 30 µmol/kg of heme, and their hepatic macrophages were evaluated 15 h after injection. Again, M1 and M2 markers were increased and decreased, respectively. The involvement of the TLR4 signaling pathway was evaluated in vitro and determined to be involved based on the attenuation of the increase of M1 polarization markers when using TAK-242, an inhibitor of TLR4. The investigators reported that M1 macrophage switching must be based on the TLR4 and ROS-producing pathways. Ultimately, the authors reported that Hpx and desferrioxamine treatment diminished these effects in both in vitro and in vivo models. Human-serum albumin, which has a low affinity for heme, also diminished the effects in both models but to a lesser degree. Furthermore, in a model of SCD mice, Hpx treatment led to an increased M2-associated metalloproteinase expression alongside similar molecules. These effects favor reduced hepatic scarring and damage, as well as reduced pro-inflammatory cytokine production that would otherwise lead to transdifferentiation of hepatic stellate cells to myofibroblasts [1].

Redinus et al. proposed a mechanism involving a similar macrophage-dependent fashion for pulmonary vascular remodeling in patients with SCD [24]. Pulmonary vascular remodeling contributes to worsened pulmonary hypertension symptoms. The authors first isolated tissue obtained from autopsies of patients with pulmonary hypertension, many of whom died from right ventricle failure. The authors noted vascular remodeling with an increased intracellular iron accumulation and expression of HO1, endothelin-1 (ET-1), and IL-6. They replicated these results in an in vivo model that used exposure to free Hb and hypoxic conditions. They found that vascular remodeling was positive for IL-6, ET-1, and HO1 expression. Via circulating macrophage knockout and by using gadolinium trichloride (GdCl3), the authors suggested that the vascular remodeling, which predisposed mice to worsened pulmonary hypertension, was dependent on perivascular macrophages. Vinchi et al. specifically discussed the M1 macrophage phenotype, and Redinus et al. proposed a novel macrophage phenotype; however, both phenotypes have overlapping factors. Further experimentation should focus on whether Hpx could be protective against the production of the suggested novel macrophage (proposed to be responsible for pulmonary vascular remodeling), similar to how it was protective against M1 phenotype switching [1,24].

Although Vinchi et al. did not extensively pursue the downstream effects of elevated IL-6 outside of the macrophage implications, Gbotosho et al. [25] reported more sensitive cardiac and serum IL-6 expression in response to heme infusion in Townes-SS mice than in healthy controls. Furthermore, they extended their study to evaluate how heme-induced IL-6 production influenced cardiac hypertrophy. Cardiac hypertrophy, commonly seen after diastolic or left ventricle dysfunction, is a serious complication of SCD. Hpx, via mitigation of IL-6 production, could play a critical role in ensuring that this signal cascade is not hyperstimulated after hemolysis. Notably, heme upregulated expression of atrial natriuretic factor (Nppa) and β-myosin heavy chain 7 (Myh7) in SS mice but not in healthy controls [25].

Overall, these results indicate that the presence Hpx and scavenging of heme creates beneficial immune conditions.

### 3.2. Hemopexin Attenuation of Deleterious Complement Activation

In addition to the modulation of cytokine production, Hpx has been shown to play a profound role in affecting the activation of the complement system, which may be critical to kidney damage in SCD renal pathophysiology. Roumenina et al. [32] showed in a clinical study of blood samples of adult patients with SCD (*n* = 106) that an overactivation of the terminal complement system was significantly greater in these patients than in healthy controls (*p* < 0.001). They reported lower levels of overactivation in patients treated with hydroxyurea, which was hypothesized to be due to the lower levels of hemolysis and, consequently, reduced heme priming of endothelial cells as complement activators. C3 deposits in endothelial cells incubated with SCD patient sera were significantly reduced with the inclusion of Hpx [32]. Poillerat et al. [26] reported in a preclinical study using phenylhydrazine (PHZ)-injected mice to model hemolysis that mice coinjected with a bolus of 100 or 500 mg/kg of Hpx alongside PHZ showed reduced renal deposits of C3b/iC3b and reduced C3b in the plasma. The authors suggested that this effect was likely due to the cytoprotection offered to glomerular endothelial cells by the addition of exogenous Hpx; however, they reported an increased expression of a marker of nephropathy, hypoxic cellular stress response protein Lcn2 (NGAL), without attenuation by Hpx treatment. Treatment with Hpx attenuated increased creatinine and urea after PHZ infusion, indicating that Hpx may offer partial protection. These results support the idea that Hpx protects against activation of the complement system; however, it should be noted that at the 500 mg/kg dose, adverse effects were noted in the mice [26]. Although the conflicting effects of Hpx on NGAL and urea or creatinine leave room for further investigation, the results regarding the complement system corroborate the findings of Merle et al. [38]. Noting C3 deposition in the kidneys of patients with SCD and mice, the authors used a similar in vivo model of PHZ-induced hemolysis to investigate the impact of Hpx on kidney injury via attenuation of complement activation. Furthermore, they found that increased C3 and hemosiderin renal deposition, as well as C3b/iC3b staining, was dependent on heme induction. Additionally, they reported that increased levels of renal injury markers NGAL and kidney injury molecule 1 (Kim-1) were dependent on C3 activation because C3^−/−^ mice showed significantly attenuated NGAL and Kim-1 expression. Similarly, in their in vitro experimentation (using human umbilical vein endothelial cells), the authors reported attenuated C3 deposition and levels of markers of C3 recruitment, including P-selectin [38].

## 4. Microvascular Stasis and Vaso-Occlusion

Microvascular stasis, known as a reduction in or blockage of blood flow, is a common symptom in patients with SCD. This event often occurs when there is an increased number of capillaries with low or blocked blood flow, also known as capillary stopped-flow. Similar to the commonly experienced vaso-occlusive crisis in SCD, microvascular stasis often leads to severe pain, as well as ischemic injury in affected organs. Triggers to both microvascular stasis and vaso-occlusion include hypoxemia, dehydration, or a change in body temperature [39]. Numerous studies indicate that the presence of cell-free Hb and heme lead to worsened microvascular stasis [6,27,33]. After a preclinical study found organ toxicity in Hpx- and Hp-depleted Townes SCD mice, Belcher et al. analyzed the effect of Hpx and Hp on microvascular stasis [6]. Townes AA (control) and SS (SCD) mice were infused via the tail vein with saline, Hp, Hpx, or both Hp and Hpx. The findings revealed that after 24, 48, and 72 h, the AA mice exhibited little to no stasis (between 0.2% and 1.6%; *p* < 0.05), whereas untreated SCD mice had 10–11% stasis after 24, 48, and 72 h (*p* < 0.01). SCD mice infused with Hpx or Hp at baseline had <1% stasis after 24 h, ~2% stasis at 48 h, and 10–11% stasis at 72 h. These temporally sensitive measurements appear to indicate that free Hb, heme, and Hp, and Hpx have a direct effect on the degree of microvascular stasis in SCD mice. To continue this analysis, the authors co-infused SS mice dorsal skin fold chambers with Hb, Hb + Albumin (which binds free heme with a low affinity), Hb + Hp, Hb + Hpx, or Hb + Hp and Hpx. At 1 h after infusion, the group that received Hb or Hb + Albumin showed approximately 32% stasis. Notably, the groups receiving co-infusions of Hb with Hp, Hpx, or both showed 12.9%, 15.8%, and 11.3% stasis, respectively (*p* < 0.01). These findings indicate that microvascular stasis is profoundly impacted by the presence of Hpx and Hp at the time of hemolysis [6].

Additionally, Vercellotti et al. [27] found similar results in an in vivo experiment using bone marrow transplants from NY1DD (an SCD genotype model) in Hpx^−/−^ and Hpx^+/+^ mice to analyze stasis. After bone marrow transfusion, when HbS was fully expressed, a heme infusion of 3.2 µmol/kg led to 33 ± 3% static vessels in Hpx^−/−^ mice compared to 21 ± 5% static vessels in Hpx^+/+^ mice (*p* < 0.025) via their dorsal fold skin chamber model. Furthermore, the authors investigated the isolated ability of Hpx to prevent stasis via the incorporation of a transposon to hepatically overexpress Hpx in Townes-AA, Townes-SS, NY1DD, and control mice. SCD model mice that received the Hpx transposon had relatively similar levels of plasma Hpx as controls, though to a varying degree by the group. In NY1DD mice, Hpx-transposon treated mice had significantly less microvascular stasis (8.9 ± 3.4%) than control injections (34.9 ± 3.4%). This pattern was similar in Townes-SS mice. Further experimentation by the authors bolstered the indication that the protective quality of Hpx, both in binding to heme as well as CD91/LRP1, likely occurred in part via induction of HO1 expression [27]. These studies provide evidence of Hpx-induced protection in sickle cell models that aids in the overall integrity of diseased vasculature and reduces the rate of stasis in SCD mice.

Whereas the prior two articles focused on microvascular stasis and vaso-occlusion of dorsal cutaneous venules, it is important to consider how Hpx may play a role in the tight arterial cerebrovasculature. Although the dynamics of immunomodulation of free Hb and heme differ slightly in the central nervous system (CNS) because of the blood–brain barrier, patients with SCD are at a greater risk for acute or chronic ischemic CNS events [40,41,42,43]. One prevalent concern is the incidence of silent stroke, in which ischemic damage could occur without any overt symptoms because of the increased basal hemolysis in patients with SCD. Bernaudin et al. reported in a longitudinal cohort of 217 newborn patients with SCA (SS/Sβ0) that 37% experienced a silent stroke before the age of 14 [44]. A systematic review and meta-analysis by Houwing et al. reported similar results, with a majority of studies showing silent cerebral infarcts in 20% to 60% of patients [45]. A recent evaluation of Hpx levels and incidence of silent cerebral infarct was conducted as part of the Silent Cerebral Infarct Transfusion Multi-Center Clinical (“SIT”) Trial. Donna Whyte-Steward, MD, reported in her contributing thesis that participants in the SIT trial (*n* = 340) with silent cerebral infarcts had significantly lower levels of Hpx than patients without silent cerebral infarcts (*p* = 0.0082). Additionally, she reported that across multivariable logistic regression analysis, Hpx was the variable most statistically associated with the incidence of silent cerebral infarct [34,46]. Future research in SCD should be directed toward evaluating low Hpx as a potential indicator for silent cerebral infarct (by initiating early and regular transcranial imaging), as well as a potential point of pharmaceutical modulation.

## 5. Heme Oxygenase-1

HO1 is a common integral membrane protein responsible for heme degradation. It plays an essential role in vascular cytoprotection and the reduction of heme-mediated oxidative toxicity. HO1 and Hpx work in combination to break down and eliminate heme, resulting in an increase in overall neuroprotection. According to Dong et al. [21], HO1 is the rate-limiting enzyme in heme degradation; therefore, it is an essential regulatory protein for the manipulation of free heme levels after injury or disease. HO1 provides cytoprotection in part through its ability to upregulate endothelial progenitor cells that aid in the regeneration of the endothelial lining of damaged blood vessels, as well as produce the anti-inflammatory molecules biliverdin and carbon monoxide, which reduce P-selectin and von Willebrand factor mobilization to cell surfaces [6,21,47,48]. This decreased mobilization plays a significant role in maintaining vascular integrity and preventing subsequent vaso-occlusion. Hpx plays a role in the induction of HO1, primarily after Hpx has already complexed with free circulating heme, though this induction may require further investigation [20,49,50]. The effects of HO1, modulated via Hpx, are particularly important in the CNS-related effects of SCD, particularly because patients with SCD are deficient in Hpx at a basal level.

It has been well-documented that the presence of the Hpx–heme complex in the CNS induces upregulation of HO1 expression. Hahl et al. [49] reported in an in vitro study using human neuroblastoma cells that the exogenous Hpx–heme complex (25 µM) induced HO1 expression, which was visualized via immunoblotting. HO1 has the ability to mount an antioxidant and anti-inflammatory response. The authors investigated whether the Hpx–heme complex was stable in ROS conditions inherent to an exacerbated inflammatory state via exposure to *tert*-butyl hydroperoxide and hydrogen peroxide. At comparable physiological levels of ROS exposure, Hpx–heme retained stability; however, the albumin–heme complex was more sensitive to denaturation. Furthermore, the authors found that Hpx–heme complexes exposed to *tert*-butyl hydroperoxide and hydrogen peroxide continued to induce HO1 expression (*p* = 0.0029 and *p* = 0.0151 at 1:1 and 1:10 molar concentrations, respectively, for *tert*-butyl hydroperoxide and *p* = 0.0067, *p* = 0.0227 at 1:1 and 1:10 molar concentrations, respectively, for hydrogen peroxide). The Hpx–heme complex likely plays a prominent role in inducing the potential neuroprotective effects of HO1 that are critical to the CNS health of patients with excessively hemolytic SCD, even in ROS conditions [49]. This conclusion is further supported by the in vivo and in vitro studies conducted by Li et al. [20]. Postnatal mouse neurons from wildtype and HO^−/−^ mice were incubated with *tert*-butyl hydroperoxide and heme. Coincubation with Hpx was cytoprotective in WT mice only; thus, the authors proposed that HO1 expression is necessary to induce a protective effect. The authors reported worse neurological deficit scores after transient ischemic experimentation in Hpx^−/−^ mice. In embryonic cortical neurons incubated with Hpx–heme, HO1 was expressed in a dose-dependent manner as observed via immunoblotting. The authors also reported that HO1 was necessary to reduce heme-induced toxicity in human embryonic kidney cells, which was exacerbated after HO1 inhibition with SnPPIX [20]. These studies suggest that HO1 induced by the Hpx–heme complex is likely profoundly beneficial in mitigating the neurodegeneration that is often associated with SCD pathology. This neurodegeneration is possibly due to the increased incidence of silent cerebral infarcts or ischemia resulting from cerebrovascular injury and vaso-occlusion.

The induction of HO1 expression in the CNS by the Hpx–heme complex has been well-identified, but there are additional pathways in the regulatory cascade of systemic HO1 expression that remain to be studied. Liu and colleagues [50] reported that HO1 expression in all monocyte subtypes observed was significantly greater in patients with SCD than in health 2000 donors (*p* < 0.001). HO1 expression was significantly lower on endothelial patrolling monocytes in patients with SCD who recently experienced vaso-occlusive events (*p* < 0.05), whereas the levels of free circulating heme were higher than in the control patients (*p* < 0.05). The authors proposed that the vaso-occlusive events may have occurred as a result of reduced basal levels of patrolling monocytes. Interestingly, in their in vitro experiments using human umbilical vein endothelial cells, heme-induced production of high HO1–expressing patrolling monocytes was blocked by the coincubation of Hpx. Similarly, while hemin induced increased HO1 expression in classical monocytes, it did not induce expression of high HO1–expressing patrolling monocytes. The authors proposed the necessity of direct endothelial exposure to heme specifically to induce increased expression of the particular high HO1–patrolling monocyte variant. Further exploration is needed on this potentially compensatory immune heme–free heme regulatory system that may be relevant for patients with SCD who are deficient in Hpx [50]. Theoretically, the Hpx clearance system is fully saturated with heme and deficient in plasma Hpx in patients with SCD; therefore, the endothelium-based system of HO1 expression induced in patrolling monocytes in circulation could be a secondary attempt at free heme degradation. The interaction between Hpx and both systemic and neurological expression of HO1 must be further studied to determine the best therapeutic approach for patients with SCD.

In total, these reported results indicate the neuroprotective role of HO1 against free heme and vaso-occlusion in SCD models, as well as the dynamic role of Hpx in vascular protection.

## 6. Hemopexin and Lipoproteins

### 6.1. Hemopexin Effects on Lipoprotein Expression

Cholesterol levels are closely monitored in patients with SCD because high blood cholesterol is a major risk factor for SCD that results in impaired endothelial function. On the other hand, low cholesterol levels also pose a risk to these patients and can lead to numerous complications, including hypocholesterolemia. Hypocholesterolemia occurs when abnormally low levels of cholesterol are found in the blood. This condition is usually found in patients with inherited or secondary diseases and blood disorders, such as SCD. SCA has been linked to a decrease in circulating cholesterol in relation to hemolysis because the demand for cholesterol in newly synthesized reticulocyte membranes increases [33]. A clinical study conducted in 2018 analyzed the relationship between lipoproteins, heme, and Hpx through cholesterol fractions in patients with SCD [35]. Study groups included 37 healthy volunteers (group AA), 39 patients with Hb-SCD (group SC), and 40 patients with homozygous SCA (group SS). Blood samples were centrifuged. Erythrocyte and Hb counts were lower in SC or SS groups than in the AA group (*p* = 0.0001), and they negatively correlated with heme and lactate dehydrogenase (LDH) levels. Additionally, low-density lipoprotein (LDL) and total cholesterol (TC) were lower in SS and SC patients than in AA patients (*p* = 0.001). TC also positively correlated with Hpx and Hb (*p* < 0.0001) [35]. Differences in the SS and SC groups showed a correlation between anemia and hemolysis with hypocholesterolemia. These findings also support the association between increased heme levels with decreased Hpx and TC levels in patients with SCD [35]. In a similar cross-sectional study, 272 children with SCD participated in testing to analyze the relationship between cholesterol, Hpx, Hp, and heme [33]. The results were similar to those of Vendrame et al. High-density lipoprotein (HDL), and LDL cholesterol levels were lower in patients with increased hemolysis (SS < SC < AA; *p* < 0.001). Additionally, Hpx was negatively correlated with LDH (r = −0.509, *p* < 0.001) and plasma heme (r = −0.592, *p* < 0.001). Hpx and Hp were also depleted in SCD groups relative to control patients. Although the study methods did not distinguish the isoenzyme being measured, results from prior studies in SCD have confirmed LDH as a reliable marker of intravascular hemolysis [33,51]. In both of the previously mentioned studies, the authors reported the significance of lipoprotein, heme, and Hpx. HDL, LDL, and TC levels were lower in patients with SCD than in AA volunteers [33,51]; therefore, there is evidence that increased hemolysis may be positively correlated with hypocholesterolemia. Future research should investigate Hpx as a way to reduce cholesterol and lipoprotein limitations on reticulocyte production (possibly packing fetal Hb via hydroxyurea) and reduce the need for exogenous transfusion to mitigate anemia.

### 6.2. Hemopexin Effects on Lipoprotein Oxidation

The aforementioned depleted levels of Hp and Hpx in patients with SCD lead to increased levels of Hb and heme, respectively, after hemolytic events and are known to increase the surrounding environment’s susceptibility to harmful oxidation. After hemolysis, free Hb is promptly oxidized from Fe^2+^ to Fe^3+^ by NO and hydrogen peroxide, which causes heme to dissociate into circulation. Circulating heme inserts itself into the core of lipoproteins, which manufactures these newly configured lipoproteins as targets for lipid peroxidation. Free radical formation from iron and hydrogen peroxide via the Fenton reaction degrades lipids, specifically in SCD, when oxidation is aggravated by elevated levels of free Hb and heme. In one study, the investigators evaluated plasma samples from patients with SCD undergoing regular transfusion therapy and found significantly lower levels of Hpx, LDL, and HDL than in controls. Additionally, significantly higher levels of free heme (*p* < 0.05 for all groups) were found. Oxidized LDL deposition was also decreased in patients with SCD, specifically in the pulmonary artery and lung endothelium [28]. These findings suggest that continuous lipid peroxidation as a result of hemolysis, as well as decreased levels of Hpx, is detrimental to SCD pathology. Moreover, another clinical study evaluated oxidative stress in non-transfused patients with SCD and reported elevated levels of metHb in affected patients, indicating an increased level of auto-oxidation of sickled Hb. Although increased levels of oxidation were found in patients with SCD, various substrates for redox cycling, including NADH and NADPH, remained relatively consistent throughout experimentation and were proposed as the reason for the lack of significant findings regarding oxidative stress [36]. This finding contrasts with earlier reports of decreased NAD in patients with SCD; there is a need for continued experimentation regarding the effects of Hpx depletion and oxidation [52].

## 7. Hemopexin and Acute Kidney Injury

In addition to the oxidative and inflammatory effects of Hpx depletion and excessive free heme, which could propagate kidney injury in SCD, a study by Ofori-Acquah et al. [29] suggested a pathway via α1-microglobulin (A1M). Although A1M, a 26-kDa protein, binds to free heme with a lesser affinity than Hpx, it plays a more significant role when Hpx is depleted. The authors reported a simultaneous 1.6-fold increase in A1M and a 4.4-fold reduction in Hpx in patients with SCD (*n* = 40) versus healthy controls. Interestingly, the A1M/Hpx ratio of these patients was directly correlated with markers of acute kidney injury (AKI), NGAL, and Kim-1. The authors proposed that this finding was due to the ability of heme-bound A1M to pass through glomerular filtration and be exposed to the renal tubule. The specific effects of A1M on AKI were tested in an in vivo model of SS and AA mice. SS mice had a 7-fold higher A1M/Hpx ratio than healthy controls, much like the human tissue samples. The authors evaluated the glomerular filtration rate (GFR), Kim-1, and NGAL and found that AKI was exacerbated in SS mice when they were challenged with a hemin injection. The role of Hpx depletion was confirmed by similar results of kidney function in an Hpx^−/−^ model as in WT mice. Fascinatingly, an injection of purified human Hpx in SS mice attenuated the decline in GFR, which is a characteristic of worsened AKI. Alternatively, A1M worsened renal dysfunction by 39% [29]. Thus, future studies of the dynamic of Hpx versus A1M heme clearance could lead to the better treatment of SCD symptoms to balance the management of renal and hepatic pathologies.

## 8. Conclusions

There is ample evidence that Hpx is a unique, multitasking protein that plays a significant role in heme-mediated protection against microvascular stasis, vaso-occlusion, and lipid oxidation, specifically in patients with SCD. The findings from this review support the argument for continued preclinical studies and possibly subsequent clinical studies regarding the use of Hpx as an intervention for SCD pathologies. Although much remains to be elucidated before suggesting a formal Hpx therapeutic regimen, the deleterious effects of free heme and the potentially cytoprotective effects of Hpx have become apparent in recent research in this field.

## Figures and Tables

**Figure 1 ijms-22-06408-f001:**
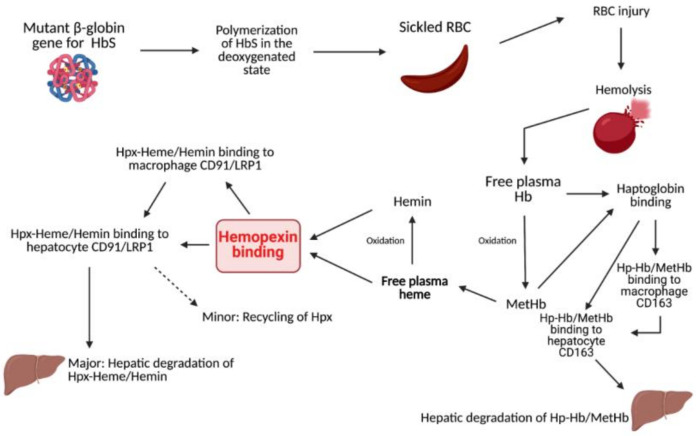
Summarization of heme and Hb clearing pathways.

**Figure 2 ijms-22-06408-f002:**
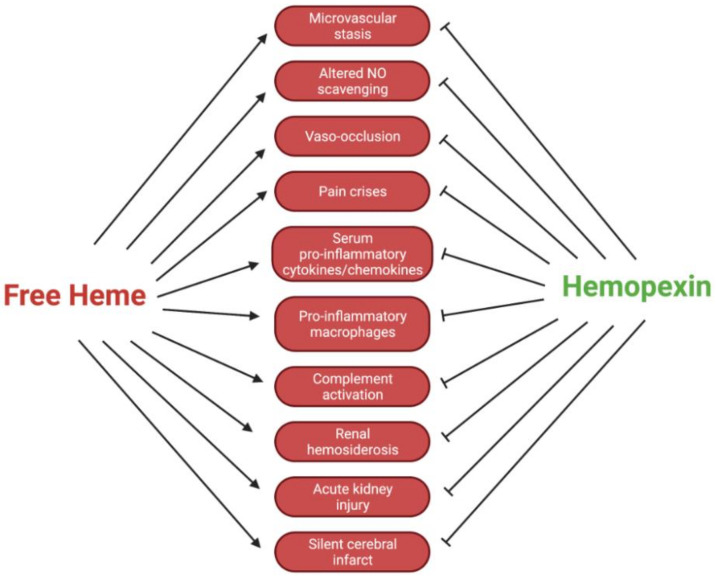
Regulatory effects of Hpx in SCD pathologies.

**Table 1 ijms-22-06408-t001:** Preclinical Hpx and SCD Research.

Reference	Age, Species, Sample, Sex, Duration	Model Used	Outcome
Vinchi, et al., 2016 [1] Hemopexin therapy reverts heme-induced pro-inflammatory phenotypic switching of macrophages in a mouse model of sickle cell disease	2–3 mo B6J & SV129 *n* = 14 15 day	Raw264.7 cells and bone-marrow derived macrophages (BMDMs) were treated with one of the following treatment groups: hemolytic aged RBCs (2 × 107), heme or Zn-mesoporphyrin (5–15 μM) bound to albumin or Hpx (5–15 μM), iron-nitrilotriacetate (FeNTA) alone (100 μM), or bound to deferoxamine (100 μM)	Hpx^−^^/^^−^ + heme mice had higher levels of HO1 (*p* < 0.05), L-Ferritin (*p* < 0.01), FPN (*p* < 0.01), and IL-6 (*p* < 0.05) than WT + heme group. In vitro treatment of Hpx + heme had lower levels of HO1 (*p* < 0.001) and Ferritin (*p* < 0.001), and FPN (*p* < 0.01) at 10 h incubation and lower levels of ROS (*p* < 0.001), TNFα (*p* < 0.01), and IL-6 (*p* < 0.01) at 15 h. BMDMs exposed to heme or iron sources showed significantly increased M1 (pro-inflammatory) macrophage polarization. This effect diminished with Hpx treatment in vivo.
Belcher, et al., 2018 [6] Haptoglobin and hemopexin inhibit vaso-occlusion and inflammation in murine sickle cell disease: Role of heme oxygenase-1 induction	12.6 ± 2.5 week, 24.0 ± 5.1 g Townes SS (*n* = 4/group) & Townes AA (*n* = 3/group) 50% M 72 h	Via tail vein, Hb was injected by itself or conjointly with saline, albumin (1 μmol/kg), Hp (1 μmol/kg), Hpx (1 μmol/kg), or Hp + Hpx (0.5 μmol Hp/kg + 0.5 μmol Hpx/kg)	Untreated AA-mice showed minimal or no stasis (*p* < 0.05) and untreated SS-mice displayed 10–11% stasis at all time points, 24, 48 and 72 h (*p* < 0.01). SS-mice treated with Hp or Hpx at baseline had <1% stasis 24 h after injection, ~2% stasis at 48 h, and 10–11% stasis at 72 h (*p* < 0.05 for all groups). SS-mice injected Hb and either Hp, Hpx, or both Hp and Hpx showed lower NF-ҡB phospho-65 levels (*p* < 0.01), displaying anti-inflammatory effects.
Redinus, et al., 2019 [24] An Hb-mediated circulating macrophage contributing to pulmonary vascular remodeling in sickle cell disease	10 week (300–350 g) Sprague-Dawley rats *n* = 58 100% M 35 day	35 mg/d Hb infusion and stimulated high altitude hypobaric Hpx (5500 m, 18,000 ft) were continued for 35 day. Hb + Hpx grouped mice were treated with 10 mg/kg gadolinium trichloride (GdCl3) 3 times a week for either 18 or 35 day	Whole lung protein quantification for HO1, IL-6, and ET-1 demonstrated an increased expression of these proteins in the Hb + Hpx cohort compared to control and GdCl3 treated Hb + Hpx groups (*p* < 0.05 for HO1; *p* < 0.01 for IL-6 and ET-1).
Gbotosho, et al., 2020 [25] HemeiInduces IL-6 and cardiac hypertrophy genes transcripts in sickle cell mice	12–16 week Townes’ knocked-in (SS), (AA), & C57BL/6J *n* = 6–10 (SS &AA) 3 h	Mice injected via tail vein with hemin (50 μm/kg/ body weight) for SS and AA mice. A hemin dose of 120 μm/kg/ body weight was used for Nrf2^+/+^ and Nrf2^−/−^ C57BL/6 mice. Control mice received sterile vehicle (0.25 M NaOH; pH 7.5 using HCl) in preparation of hemin	IL-6 and Hmox1 expression increased in Townes SS mice compared to AA controls (*p* ≤ 0.05 and *p* ≤ 0.01, respectively). After heme injection, SS mice showed a 15.4-fold increase in cardiac IL-6 transcripts compared to the vehicle control group (*p* ≤ 0.05) and roughly a 53% increase when compared to AA control mice (*p* ≤ 0.001). 3 h after heme treatment, SS mice showed an increase in Nppa (14.8-fold; *p* ≤ 0.001) and Myh7 (8.1-fold; *p* ≤ 0.01) transcripts, while AA control mice showed no increase. In heme-treated Nrf2^−/−^ and Nrf2^+/+^ mice, cardiac IL-6 was 51% higher in the Nrf2^−/−^ group (*p* < 0.01).
Poillerat, et al., 2020 [26] Hemopexin as an inhibitor of hemolysis-induced complement activation	8 week (0.125 mg/g) C56BL/6 *n* = 10 (Hpx injected) *n* = 8 (control) 100% M 72 h	3 groups of mice were injected IV with 100 or 500 mg/kg of either human plasma derived Hpx (CSL Behring) or PBS. After this, mice were immediately injected IP with phenylhydrazine (PHZ) at a concentration of 0.125 mg/g body weight. Control mice received PBS injections at an equivalent volume of Hpx and PHZ	Clearance of plasma heme by Hpx was 4-fold higher (100 mg/kg group) and 2-fold higher (500 mg/kg group) in mice treated with PHZ than controls. Kidney injury decreased in Hpx pretreated PHZ-infused mice. At 6 h, urea measurements were decreased in 100 mg/kg Hpx mice (*p* < 0.005) and creatinine measurements were decreased in both 100 and 500 mg/kg Hpx mice (*p* < 0.05 and *p* < 0.001, respectively) compared to PBS-infused controls. PHZ-induced deposits of C3b/iC3b were attenuated in both 100 and 500 mg/kg Hpx groups (no *p*-values reported).
Vercellotti, et al., 2016 [27] Hepatic overexpression of hemopexin inhibits inflammation and vascular stasis in murine models of sickle cell disease	8–20 week (20–30 g) NY1DD & Townes AA & SS (*n* = 4 for all groups) 50% M 12 week	High-pressure injection of DNA solution (constructed via plasmid manipulation) into tail vein (10% body weight, up to 2.5 mL). Plasmids were either WT-Hpx, missense-Hpx, or Luc plasmid	1 h after infusion with heme, Hpx^−/−^ sickle mice developed 33 ± 3% static vessels compared to 21 ± 5% in control sickle mice (*p* < 0.025).
Yalamanoglu, et al., 2018 [28] Depletion of haptoglobin and hemopexin promote hemoglobin-mediated lipoprotein oxidation in sickle cell disease	20 week C57BL/6 (*n* = 5), BERK-SS (*n* = 9), & Hartley guinea pigs (GP) (*n* = 8) 50% M 2 week	PE-50 tubing catheters were placed into the left external jugular vein of GPs. High diet (HD) GPs (*n* = 4) were infused with heme-albumin (5 and 25 mg/kg doses) while another group of HD GPs (*n* = 4) was dosed with vehicle (0.9% NaCl) via the jugular catheter. Blood samples taken from mice were centrifuged at 2000 rpm for 15 min	BERK-SS mice reveal greater formation of lipid oxidation end products, malondialdehyde (MDA), compared to WT mice. Specifically, HDL and LDL concentrations were 0.3 ± 0.1 μm/80 μg and 0.3 ± 0.4 μm/80 μg (*p* < 0.05 for both groups). Similar MDA levels were seen in HD GP groups sampled 8 h after infusion for GPs injected with either 5 or 25 mg/kg/body weight heme-albumin (*p* < 0.05).
Ofori-Acquah, et al., 2020 [29] Hemopexin deficiency promotes acute kidney injury in sickle cell disease	20 μmol/kg Knockin Townes SS & AA *n* = 8–12 (SS) & 8–12 (AA) 2 day	Hpx^−/−^ and C57BL/6 mice were transplanted with SS bone marrow. Heme infusion (20 μmol/kg) mentioned, although location not specified	After heme infusion, excess heme was preferentially transported to the kidneys in SS mice (*p* < 0.01) versus the liver in AA mice (*p* < 0.05). Additionally, in SS mice, plasma creatine (Cr) and urinary albumin-Cr ratio increased 4-fold (*p* < 0.001) and 2-fold (*p* < 0.01), respectively, compared to baseline levels. Levels of α-1-microglobulin (A1M) were elevated in SS mice compared to controls (*p* < 0.001) while Hpx levels were low in both groups. The Hpx^−/−^ SS mice had the lowest baseline glomerular filtration rate (*p* < 0.05) and increased kidney injury molecule-1 (KIM-1) compared with Hpx^−/−^ and control mice (*p* < 0.001).
Chintagari, et al., 2015 [30] Haptoglobin attenuates hemoglobin-induced heme oxygenase-1 in renal proximal tubule cells and kidneys of a mouse model of sickle cell disease	8–12 mo (20–30 g) NY1DD mice *n* = 4 7 day	SCD mice were infused via the tail vein with one bolus of Hb (3.2 μm/kg), Hp (3.2 μm/kg), or Hb-Hp complex (0.012 mL/g)	Hp attenuated Hb-induced HO1 mRNA expression in SCD mice at 4 h after Hb infusion (*p* < 0.05). Additionally, analyzed via Western blot, Hp attenuated the Hb-induced expression of HO1 protein in the proximal tubule of the kidney of SCD mice.
Camus, et al., 2015 [31] Circulating cell membrane microparticles transfer heme to endothelial cells and trigger vasoocclusions in sickle cell disease	10–18 week C57BL/6J mice *n* = 6 100% M	Microparticles (MP) were purified and then injected via IV administration	MP injection increased circulating erythrocyte MP concentration by 25% and reduced both diastolic and mean blood flow velocity in the renal arteries by 30%, within 5 min (*p* = 0.009). Heart rate and cardiac output were not affected after injection of MPs. Mice treated with MPs that were pretreated with 1µM Hpx had no change in blow flow velocity (*p* = 0.72).

**Table 2 ijms-22-06408-t002:** Clinical Hpx and SCD Research.

Reference	Age, Sample, Sex, Duration, Location	Study Design	Outcomes
Yalamanoglu, et al., 2018 [28] Depletion of haptoglobin and hemopexin promote hemoglobin-mediated lipoprotein oxidation in sickle cell disease	≥18 year *n* = 19 (SCD) & *n* = 11 (controls) 50% M Aurora, CO, USA	Prospective Case-control study	In SCD patients, HDL and LDL plasma levels were lower compared to controls (25 ± 16 and 82 ± 26 mg/dl vs control 53 ± 18 and 128 ± 21 mg/dl). SCD patients had an increase in heme bound to Hb (9.5 ± 4 μM) and total plasma heme levels (60 ± 23 μM) compared to plasma obtained from control patients (4.6 ± 4.2 μM and 19 ± 11 μM).
Camus, et al., 2015 [31] Circulating cell membrane microparticles transfer heme to endothelial cells and trigger vasoocclusions in sickle cell disease	≥18 year *n* = 47 (SCD), *n* = 22 (AA) Paris, France	Retrospective Case-control	In patients with SCD, annexin-a5+ MPs were increased 5-fold and plasma Hb were increased 3-fold compared to controls (*p* < 0.001 and *p* = 0.007, respectively). SCD erythrocytes released MPs that contained 2X the concentration of heme compared to control MPs (*p* = 0.038).
Roumenina, et al., 2020 [32] Complement activation in sickle cell disease: Dependence on cell density, hemolysis and modulation by hydroxyurea therapy	≥18 year *n* = 106 (SCD) & *n* = 176 (AA) Paris, France	Retrospective	sC5b-9, a marker of complement activation, was increased in 42% of SCD patient plasma (*p* < 0.001) and was higher in SCD patients treated with hydroxyurea compared to the nontreatment SCD group (*p* < 0.005). In nontreated-SCD patients, sC5b-9 levels were higher after incubation of SCD RBCs with normal human serum compared hydroxyurea-treated SCD and control RBCs (*p* = 0.038 and *p* = 0.006, respectively). Endothelial cells incubated with SCD serum led to the C3 deposits on the surface of cells and were decreased after the addition of Hpx (*p* = 0.0117).
Santiago, et al., 2018 [33] Serum haptoglobin and hemopexin levels are depleted in pediatric sickle cell disease patients	<18 year *n* = 179 (SS), *n* = 93 (SC), *n* = 28 (AA) University of Bahia, Brazil	Cross-sectional study	SS and SC patient groups showed lower Hp and Hpx levels (SS < SC < AA) and increased reticulocyte counts and serum LDH (SS > SC > AA). SCD patient (SS and SC) serum analysis showed a negative correlation between Hpx and LDH (r = −0.509, *p* < 0.001) and heme (r = −0.592, *p* < 0.001).
Whyte-Stewart, et al., 2017 [34] The association between hemopexin and silent cerebral infarcts in pediatric sickle cell disease	5–15 year *n* = 340 Baltimore, Maryland	Prospective Cross-sectional study	In patients who experienced silent cerebral infarcts, Hpx levels were lower than controls (*p* = 0.0082). Hpx was shown to be associated with an increased risk of silent cerebral infarct (*p* = 0.03).
Vendrame, et al., 2018 [35] Differences in heme and hemopexin content in lipoproteins from patients with sickle cell disease	35 year (AA; *n* = 37; 32% F), 45 year (SC; *n* = 39; 62% F), 37 year (SS; *n* = 40; 48% F) Campinas, Brazil & Seattle, WA, USA	Prospective Case-control study	In SCD groups (SC and SS) compared to controls (AA), erythrocytes and Hb concentrations were lower while reticulocyte, bilirubin, and serum LDH levels were increased (*p* = 0.0001). In SS group alone, heme was reported at higher levels compared to AA or SC groups (*p* = 0.0001). In SC and SS patient groups, soluble VCAM-1 levels were increased compared to controls (*p* = 0.0001) and were shown to be higher in SS patients over SC patients (*p* = 0.02). Additionally, in SS and SC groups, total cholesterol and LDL levels were decreased compared to controls (*p* = 0.001).
Detterich, et al., 2019 [36] Erythrocyte and plasma oxidative stress appears to be compensated in patients with sickle cell disease during a period of relative health, despite the presence of known oxidative agents	23 ± 8.8 year (SCD; *n* = 55) 27 ± 8.9 year (AA; *n* = 44) 28 day Los Angeles, California	Retrospective Case-control	Measured at the time of phlebotomy, SCD patients demonstrated higher % metHb (*p* = 0.004). In the plasma, free Hb and heme, were elevated in SCD patients compared to controls and increased rapidly at Hpx levels less than 200 μg/mL SCD patients.
Santaterra, et al., 2020 [37] Endothelial barrier integrity is disrupted in vitro by heme and by serum from sickle cell disease patients	>18 year *n* = 20 (SCD), *n* = 10 (AA) Campinas, Brazil	Retrospective Case-control In vitro study	In SCD patients, Hpx levels were lower than comparative healthy volunteers (0.33 ± 0.32 vs 1.29 ± 0.23; *p* < 0.001). Heme-induced endothelial barrier disruption was correlated with Hpx levels, most significantly at 12 min (R_s_ = 0.68; *p* < 0.0001).

## Data Availability

Not applicable.

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
