# Peer review of "Critical Role of Hemopexin Mediated Cytoprotection in the Pathophysiology of Sickle Cell Disease"

_ijms, 2021, doi:10.3390/ijms22126408_

Round 1

Reviewer 1 Report

This is a nice and compact review of the physiological aspects of hemopexin and its potential pharmaceutical use.  It's clear and easy to follow.

My suggestions:

  1. Since planning to appear in "Int. J. Molecular Sciences" , I think the molecular/structural side of the topic should also be presented in the paper.  Two crystal structures of rabbit hemopexin (1qjs, 1qhu) and a newer one from grass pea (3lp9) are available from the Protein Data Bank.  At least the former should be presented and discussed.
  2. There is also a paper in Biol. Chem. (Revisiting the interaction of heme with hemopexin; https://www.degruyter.com/document/doi/10.1515/hsz-2020-0347/) discussing the homology modelling of human hemopexin and the possibility of multiple heme-binding sites vs the accepted 1:1 association - this could also be referenced
  3. In Table1 1st column: reference numbers should be superscripted
  4. The "2. Search terms and strategy" chapter is unnecessary.

Reviewer 2 Report

General comment:

This review, entitled “Critical role of hemopexin mediated cytoprotection in the pathophysiology of sickle cell disease,” authored by Ashouri et al., reports comprehensive updates on current research regarding the dynamic activity of hemopexin implicated in sickle cell disease. This review also includes in vivo studies in mouse, rat, and guinea pig models of sickle cell disease and studies in human samples. This is a kind of review suitable for identifying the role of a particular protein in pathological conditions. In my opinion, this review is suitable for publication in Int J Mol Sci after the authors have addressed the following comments and questions::

Specific comments:

  • Hpx-heme complexes identified by macrophage through membrane-bound receptor CD91. What is the role of this receptor in hepatocytes? Can it help in Hpx-heme complexes identification by hepatocytes?
  • The role of heme oxygenase I in heme and Hb clearing pathways should be elaborate?
  • Sickle Cell Disease in a combination of beta-thalassemia should be mentioned-there is scope in this review - paragraph 1.2.
  • How the haptoglobin, hemopexin, and HO1 are regulated in heme and Hb clearing pathways to avoid heme toxicity?
  • The inflammatory cascades triggered by toll-like receptors -2 and -4, primarily activate innate immunity. How hemopexin and haptoglobin-related research will be helpful in immunosuppressive therapies.
  • Hemoglobin bound to haptoglobin is capable of oxygen binding. Is that true for hemopexin-bound heme too? Is there information available for heme iron state in the Hpx-heme complex?
